# Current Status and Conservation Management of Farm Animal Genetic Resources in Bhutan

**DOI:** 10.3390/vetsci10040281

**Published:** 2023-04-06

**Authors:** Jigme Tenzin, Vibuntita Chankitisakul, Wuttigrai Boonkum

**Affiliations:** 1Department of Animal Science, College of Natural Resources, Royal University of Bhutan, Punakha 14001, Bhutan; jtenzin.cnr@rub.edu.bt; 2Department of Animal Science, Faculty of Agriculture, Khon Kaen University, Khon Kaen 40002, Thailand; vibuch@kku.ac.th; 3Network Center for Animal Breeding and Omics Research, Faculty of Agriculture, Khon Kaen University, Khon Kaen 40002, Thailand

**Keywords:** in situ, ex situ, conservation, genetic diversity

## Abstract

**Simple Summary:**

The diversity of animal gene pools in terms of species and numbers indicates a country’s food security. In the past, Bhutan was very rich in livestock resources. However, if there is no plan to support the conservation and utilization of livestock resources well, it may inevitably cause food shortage problems. The main message of this paper is that some animal species are showing a steady decline in their numbers without the cooperation of many organizations, and the lack of knowledge, in particular the genetic improvement and breeding selection, for maintaining the integrity of the country’s available animal genetic resources may soon enter a crisis.

**Abstract:**

Farm Animal Genetic Resources (FAnGR) ensures food security and maintains genetic diversity. The efforts to conserve FAnGR are minimal in Bhutan. In the pursuit of increasing livestock productivity, farmers are rearing livestock that narrows the range of genetic diversity. This review attempts to summarize the status of FAnGR and the efforts to conserve them. Some unique livestock breeds in Bhutan are Nublang (cattle breed), Yak, Saphak (pig breed), Yuta (horse breed), Merak-Saktenpa (horse breed), and Belochem (chicken breed). There was a drop in the yak, buffalo, equine, pig, sheep, and goat populations. Both in situ and ex situ conservation measures are in place for some of the breeds and strains (e.g., Nublang and traditional chicken). Conservation efforts are limited to the government, but other individuals, stakeholders and non-government organizations must play an increasing role in preserving genetic diversity. It is pertinent that Bhutan comes up with a policy framework to conserve its indigenous cattle.

## 1. Introduction

Presently, the Food and Agriculture Organization’s (FAO) Domestic Animal Diversity Information System (DAD-IS) reported that about 8800 different livestock breeds exist, which is often referred as “global animal genetic resources (AnGR)” with about 38 different species to meet our needs for food, clothing, draft power, and saving money [1]. These animal species, breeds, and strains have economic, scientific, and cultural values to mankind for food and agricultural production [2,3] playing a crucial role in global food systems as the main source of animal protein, and one third of all protein consumed by people comes from animal-source foods [4]. Especially, native breeds are well adapted to local agro-ecological zones and resistant to emerging diseases [5] and provide sustained economic returns albeit small.

The preservation of indigenous animal breeds is essential because with intense selective breeding for commercial purposes, the genetic variability has been decreased both within and across strains [6], leading to the extinction of native animals with as much as 28.83% of local breeds being reported at risk because of genetic erosion from government programs and policies as well as climate change in terms of heat stress [4,7]. This is especially true for developing countries such as Bhutan. For instance, the extension of markets and economic globalization has significantly contributed to the loss of breeds [8] affecting the local livestock diversity in the country. Therefore, the conservation of livestock diversity is essential [9]. According to Boettcher et al. [6], many native breeds have been lost, replaced, or genetically diluted due to a number of circumstances. For instance, farmers in developing nations are under great pressure to transition to commercialized livestock production and breeding systems, which is the major cause of genetic loss [10]. Because of agricultural policies that encourage quick fixes to ensure food security or meet rising food demand [2], crossbreeding and, as a result, the replacement of locally adapted breeds by a narrow of high-yielding international trans-boundary breeds is becoming a source of concern in developing countries, threatening animal genetic diversity [6,11].

Bhutan is a developing country located in the eastern Himalayas between China and India. In Bhutan, livestock is a critical factor that makes it possible for the people to prosper in a relatively infertile mountain environment [12]. Indigenous breeds are less productive, but it is vital to conserve them [13] as they contribute to livestock diversity in the fragile mountain ecosystem with socio-economic, religious, and cultural significance [12,14]. Moreover, they are highly acclimatized and adapted to local climate in addition to being resistant against endemic diseases and in some cases resistant to emerging diseases. The livestock diversity in Bhutan is similar to those occurring in the Himalayan region, but some breeds are unique to Bhutan. For example, Nublang (*Bos indicus*), a Siri type of breed, is considered as native to Bhutan [2]. In addition, local breeds of domestic animal present are Mithun (*Bos frontalis*), Yak (*Bos grunniens*), Sheep (*Ovis aries*), Goat (*Capra hircus*), Buffalo (*Bubalus bubalis*), Pig (*Sus domesticus*), Chicken (*Gallus domesticus*), and Horses (*Equus caballus*)—which places an important economic role in enhancing the livelihood of rural poor and farmers in the country. Figure 1 shows the available AnGR in Bhutan, including the exotic breed (*Bos taurus*) [2]. The animal genetic resources of a country are a critical factor in realizing the goal of nutrition and food security and upliftment and development of the rural communities [15]. Therefore, it is imperative that native animal germplasm be conserved.

In Bhutan, conservation of livestock diversity is a national program with the National Biodiversity Centre (NBC) [16], serving as the national focal point and erstwhile the Ministry of Agriculture and Forest (MoAF), now the Ministry of Agriculture and Livestock (MoAL), as the national competent authority for conservation of fauna and flora for conservation and sustainable use [17]. There is a dedicated Animal Genetic Resources (AnGR) program for both in situ (conservation in wild and natural habit) and ex situ conservation is in place [17]. There are also many farms established such as the National Nublang Breeding Farm (NNBF), Tashiyangphu, which was established by Department of Livestock, Ministry of Agriculture, and Forest (DoL, MoAF). In addition, cryoconservation and Artificial Insemination (AI) services of Nublang and provision of semen have been supported by National Dairy Development Centre (NDDC), Yusipang, Thimphu. However, the trend in the population growth of cattle is decreasing: a total of 309,277 cattle heads were reported in 2010, and it decreased to 281,015 in 2020, an about 10% decrease (Table 1). This might be due to a limitation on many aspects in Bhutan mainly in effective tools and advanced technology for genetic improvement and conservation.

The AnGR conservation will help us sustainably breed, select, and produce more [6]. This, in turn, ensures long-term food security as livestock are intricately linked with food and nutrition security in the developing world. The livestock agriculture in developing countries are typically characterized by low inputs. Therefore, to conserve AnGR in these countries requires taking a multifaceted approach and including as many breeds as possible that are adaptive and resistant to harsh climate and diseases [13]. Therefore, this review attempts to study the status of the farm animal diversity and conservation efforts in Bhutan. For this particular review, the term livestock refers to farmed domestic animal including poultry for convenience of discussion.

## 2. Current State of Farm Animal Genetic Resources in Bhutan

In Bhutan, the Department of Livestock, MoAL is the apex body which oversees the livestock development. The department considers the diversity of bovines, caprine, ovine, equines, avian, swine, canines, and felines [17,18]. However, in this review, we included bovines, caprine, ovine, equines, avian, and swine, which are subject of conservation interest and have economic implications (Figure 1).

### 2.1. Nublang (Bos indicus)

Cattle are among the most important domestic animal in Bhutan, providing milk, meat, draught power and manure for agriculture in the country [19,20,21]. Dorji et al. [20] classified local cattle into Nublang/Thrabam (Siri), Jaba/Guree, and Bajo/Goleng. The Nublang is the male native cattle while Thrabum is the female counterpart [19,20,21]. Jaba is a name given to cattle originating from across the border of India and classified into three types: Eastern, South Bhutan (Sarpang), and Guree (Southwest Bhutan). The Jaba type forms a smaller portion of the gene pool in Northeastern India, and their origin is probably from Assam Duars [22]. Bajo and Goleng are *Bos taurus* cattle originating form Tibet and are found along the Himalayan ranges of Laya and Lingshi [23].

Among the cattle, the Nublang (*Bos indicus*) are considered to be most important [22] and are critical for conservation (Table 2). They play a central role in nutrition and food security in rural areas of Bhutan [24]. Local legend says that the Nublang origin is linked to a lake (Nub Tshonapata), located in the mountains of Nakha village, of Sombaykha Gewog of Haa, Bhutan [19]. However, Payne and Hodges [25] reported the origin of these animals as a stabilized cross between Indian zebu and Tibetan Taurine. Nublang is divided into four groups: western, south, central, and eastern groups, of which the eastern Nublangs are distinct and similar to Indian Jaba (small cattle found in border towns of Indian states) [20]. Genetic distance and the coefficient of differentiation among the Nublang west, central, and south were reported to be the least while the Nublang east were most distinct and closer to Indian Jaba. The east Nublang have been bred with Mithun and also with Indian Jaba, which explains their distinctiveness and closeness with Indian Jaba [20]. The average heterozygosity for the genetic variability ranged from 24 to 40% in cattle in a study of a milk protein [19]. Currently, most of the population is found in Sombaykha, Gakiling, Dungtoe, and Chaling regions in Bhutan.

Distinguishing features of Nublang/Thrabum are the typical and prominent cervical hump of the *Bos indicus* type [19,26]. They have a long head and face with a wide and flat forehead. Their horns are short, sharp, and curve outward and forward—sometimes upward or inward. There are tufts of hair on polls as well as on the base of the horns. Humps are prominently developed in bulls compared to cows. Feet are long; their necks are broad with well-developed dewlap in males extending from mid-jaw to brisket [26]. They could come in any color, such as piebald and red, but a black color was reported to be most common [27].

The animals are used for milk as well as for draught power. The average milk yield per day was reported to be 2.12 ± 0.7 L from a National Nublang farm, Tashiyangphu. The average lactation milk yield was 519 ± 151 L with a lactation length of 239 days. The heritability of these two traits were estimated to be 0.22 ± 0.16 and 0.13 ± 0.12, respectively [24]. Another study in a rural setting reported the average milk yield per day of 1.52 ± 0.13 L [21]. Feeding practices in the government and rural settings might have played a role in having disparate milk production. The age at first calving for Thrabam is around five years with a lactation length of 280 days which is lower than the standard 305 days for European cattle [26].

However, as presented in Table 1, the population of cattle is decreasing. Bhutanese farmers rear four to six heads of cattle with prominent breeds being Thrabam (36%), Jersey and Brown Swiss (26% and 36%), and a few Mithun crosses (2%). Mainly the local animals are reared for self-consumption and manure, and many farmers practice free grazing management: the only cows with calf at foot return to the cowsheds, whereas the rest of the herd can also remain free in the wild [26]. Moreover, the herd is usually kept in a nearby forest away from settlements especially the local cattle [21].

Famers avail artificial insemination services if the extension center is nearby. However, natural service is practiced as well with many farmers managing their own breeding bulls (one or two in a herd). If herders do not have bulls, they will hire from others with fees being paid: if a pure Mithun is used, Nu. 500 (Nu = Bhutan’s currency) is paid for females (Jatsham) born and Nu. 400 for male (Jatsha) calves. A mencha (75:25 Mithun-Siri cross bull) is also used, and Nu. 200 is paid for Jatsham and Nu. 100 for Jatsha born. In addition, it is customary to provide mustard oil, a bottle of local wine, and eggs to the owner of the breeding bulls. Other types of bulls are provided free of charge [26].

While conventional farming with local breeds remains a phenomenon in the rural areas, the urban–periurban region is increasingly seeing a switch to higher producing Jersey and Brown Swiss–Thrabam crosses because of higher performance and better access to market, which is also restructuring the traditional herd. However, in rural areas Mithun–Siri crosses are still maintained due to their relatively higher milk yield and milk fat than other non-descript animals, and they provide greater power for draft work, and they are also highly adapted to the local environment [26].

The conservation efforts are led by research centers, breeding farms, and the regional livestock development center, the National Nublang Breeding Centre, Tashi Yangphu, Wamrong [28]. Free extension service is also provided by the Department of Livestock through the extension centers. However, labor, feed shortages, and the availability of exotic breeds are driving the local cattle population down with many giving up farming and and not incentivizing the youth to venture into farming [26].

### 2.2. Mithun (Bos frontalis)

Mithun is indigenous to the northeastern hills of India and is commonly found in Arunachal Pradesh, Nagaland, Manipur, and Mizoram. The animal is also found in Bhutan, Myanmar, Bangladesh, and China [29]. While in India it is reared for its meat [30], in Bhutan Mithun plays a significant role especially in providing Mithun–Siri crosses (known as Jatsha for males and Jatshamfor females) [21] and usually is bred in the eastern part of Bhutan, and only bulls are maintained as Mithuns are difficult to handle [22].

There is a prominent dorsal ridge on the crest of the shoulder, with a flat forehead and big horns with large base. The animals are brownish black and piebald, and most of the Mithun have white stockings. Sequencing by Dorji et al. [19] revealed that the Mithun has a low nucleotide diversity and is phylogenetically related to Gaur. A comparison with wild Gaur specimens from three locations in Bhutan proved the close kinship between Bhutanese Mithun and Gaur—supporting the domestication of Mithun from Gaur. Moreover, haplotypes of 12 Mithuns from Bhutan had Guar haplotypes based on mtDNA which suggested that the origin of Mithun is Gaur [30]. This is also further corroborated by Winter et al. [31].

In Bhutan, Mithuns (males) are crossbred with Thrabams (females of Nublang) to produce superior Jatsa and Jatsham than their parents. While Jatsa is predominantly used as a draught breed in rural areas, Jatsham is used for milk purposes. The male F1 progeny of Mithun and Thrabum is sterile while the female progeny is fertile. Therefore, as Jatsha are not fertile, they backcrossed across five generations to get a fertile bull: Jatsham with Nublang in subsequent generations to produce Yangka (male) and Yangkum (female) which the males are mostly sterile, and females are fertile. The Yangkum (female) is now backcrossed with Nublang to produce Deob (male) and Deobam (female), whereby it is shown that the males are regaining fertility. Moreover, the Deobam is crossed with Nublang to produce Deothra (male) and Deothram (female) with males regaining fertility. At the fifth generation of Deothram crossing with Nublang, a male (Nublang) and female (Thrabam) are produced with the male gaining fertility. The backcrossing for three to four generations is to regain fertility and generate replacement stocks for Nublang. The breeding schematics are presented in Table 3. How does this work? A study of spermatogenesis showed that Jatsha has no mature spermatozoa, but subsequent backcrossing has spermatozoa but in fewer numbers, but elongated spermatids were reported to be abundant [32]. Additionally, the researchers noted that gradual regaining of fertility is expected due to gene dilution and progressive segregation of genes responsible for infertility.

Mithuns are particularly important in ethnic communities; farmers highly value and consider a Mithun bull as a gem. A particular good breeding bull is equal to ‘half the herd’ with a popular belief that a calf inherits Mithun’s good qualities. Therefore, a herder pays a high price to own one in their herd. If a household owns a bull for the first time, a ceremonial procession is performed with burning incense. A khadar (white scarf) is offered, and a Marchang (wine offering ceremony) ceremony is performed followed by tea and alcoholic drinks. However, the shift from subsistence to commercial farming through Mithun cattle hybridization is threatening the Mithun population including alternative livelihood options [33]. The Mithun population has decreased by more than 30% in the last decade with farmers opting for alternative livelihood options and opportunities and commercial farming (Table 3).

Conservation efforts of Mithun are being undertaken: a Regional Mithun Breeding Farm (RMBF) is established at Arong, Samdrup Jongkhar, in the eastern part of Bhutan. Farms are mandated to breed and provide draft power to agriculture production in the rural areas and to ensure that they maintain a pure Mithun breed [34]. Another RMBF was located Zhemgang with a mandate to produce, procure, and distribute purebred Mithun in West, Central, and East Central Dzongkhags which is now combined with Arong RMBF. It has a mission to conserve and contribute to the sustainable utilization of Mithun and to develop pure line of Mithun for all times to come [35].

### 2.3. Yak (Bos grunniens)

Yaks in Bhutan are major source of livelihood for people in the highland (Brokpas and Zhops in Merak, Tashigang, and parts of Bumthang entirely depend on yak, while agropastoralists in Haa and Gasa depend on yak partially)—mostly in the alpine rangelands above 3000 masl. They provide livelihood to farmers with limited opportunities and where there are poor social and physical infrastructures [36].

Bhutanese yaks resemble those found in the Himalayas and on the Tibetan Plateau. According to genetic distance and the use of microsatellite markers, Bhutan’s yaks are grouped into two categories: west and central region yaks and eastern yaks with a unique gene pool with more genetic variation [37,38]. The western yak is larger compared to central and eastern region (Table 4).

The production of daily milk in average is 1 kg with an average lactation length of 234 days and a lactation yield of 153 kg [39]. The yak milk is converted into hard cheese called *Chugo* (Bumthang and Sephu), *Hapiruto* (hard cheese) (Haa and Paro), and *Tachu* (Lingshi, Thimphu). Fermented cheese (*Yoeshey*/*Zeotey*) is a specialty from the Merak-Sakten, Trashigang in eastern Bhutan. Moreover, yak hair and wool, meat, and hide are sold as a livelihood option [36]. The details of production and reproduction traits are presented in Table 5.

Hybridization between yaks and cattle is prevalent, resulting in strains like Zo and Zom. Another breed, Goleng, is a *Bos taurus* cattle breed from Tibet that is frequently crossed with yak [37]. However, there are several constraints to yak farming: shortages of fodder, yak mortality due to gid disease, predation, weed infestation of rangelands, inbreeding, and labor shortages. Moreover, the milk production for yak is low [39]. The same study recommends collaborating among stakeholders and reducing challenges and barriers and providing opportunities for yak farmers. Therefore, the Department of Livestock through National Highland Research Development Centre (NHRDC) is actively involved in enhancing the livelihood through many “highland program” [41]. These in turn are thought to rejuvenate the yak farming community in the country.

The population of yaks appears to have increased slightly (Table 1), which can be attributed to aggressive government policies in promoting the livelihood of highlanders. For example, in 2022, the Department of Livestock initiated a free supply of feed supplements to yaks for vulnerable yak populations as there was a severe shortage of feed [42]. Moreover, Bhutan has started the highland festival to promote the sustainable livelihood of highlanders, showcasing highlander’s innovation and culture and exhibiting the highlands as a “Pride of Bhutan” [43]. Recently, the Department of Livestock [DoL] [44] established a yak federation to promote yak farming in the country: the federation helps in promoting nutrition and health of the yaks. Moreover, they will help in identification and standardization of yak products in the country.

### 2.4. Swine (Sus domesticus)

Bhutanese people rear pigs for cultural and socio-economic reasons: owning a pig and slaughtering conveyed social status, wealth, and power in olden days [45]. The indigenous pigs are known by the names: Dompha, Dromphak, Saphak, and Jituphap from the Food and Agriculture Organization’s (FAO) Domestic Animal Diversity Information System (DAD-IS) [4]. The pig population in Bhutan is generally described as non-descript; they are similar to pigs found in West Bengal in India [45]. The mtDNA sequence study of domestic pigs and wild pigs suggested that the Bhutanese pig originated probably from Tibet or China, Southeast Asia, or East Indian wild boars [46].

Nidup et al. [45] describes four different types according to regions of Bhutan: Eastern, East–Central, Western, and West–Central pigs. The description of the local pigs is provided in Table 6. In the Sarpang district, the pigs are usually smuggled through the border and known as “Machay Sunggur” after a tribe of neighboring states of Assam. In Chukha, similar pigs are smuggled through West Bengal, India, and go by the name “Machay Madhuri”.

A study on local and exotic breeds reports that the local (Saphak) pigs’ average daily gain was 0.244 kg while exotic breeds had 0.608 kg gain. The average final market weight was 45 kg while it was 101.75 kg for exotic breeds (Saddleback) [47]. Dompha can attain up to 90–100 kg, and they mature very late. The local pigs are reared in stall floor made from local bamboos and poles which may be thatched partially. They can be free range as well as confined with tethering [48]. The production traits are given in Table 7. There are opportunities for selective breeding and promoting them for meat purpose.

In 2011, 68% of the pig population were still indigenous [45]. However, in 2020 the population of the pigs decreased by more than 10% compared to 2010 (Table 1)—which is both local and exotic breeds. Similarly, Nidup et al. [45] note the alarming loss of population of indigenous population over the years: in 1998, there were more than 85,932 pigs; in 2008, it was reduced to 16,959. Moreover, the native pig population is decreasing because the breeding policy recommends using exotic pig breeds such as Yorkshire, Duroc-Jersey, Large Black, and Saddleback [48].

A major constraint in pig farming is religious sentiments against slaughter and a lack of pig feed at a reasonable price. However, the traditional breeds are fed and managed in a conventional way—making it advantageous in reducing the cost of rearing, but with poor growth. Therefore, improvement on breeding and nutrition can enhance the management of the traditional piggery [48] that can lead to sustainable utilization and conservation of pig genetic resources. Moreover, providing incentives or subsidy through policy intervention might help farmers to rear native pigs helping their conservation.

### 2.5. Sheep (Ovis aries)

Sheep represent important genetic resources of the country. They are mainly kept for manure, mutton, social security, and wool. Wool is generally catered to domestic small-scale industries. The sheep are named after the areas they are found Jakar type, Sakten type, Sipsoo Type, and Sarpang type (Table 8). They are adapted to harsh and rugged terrains of Bhutan [49].

The sheep found in Wangdue, Trongsa, and Bumthang are similar and smaller in size, whereas sheep found in Sakten are bigger. The Sarpang type is also smaller in size. The sheep genetics showed higher genetic variation in the Jakar and Sakten types then than in the Tsirang, Sipsu, and Sarpang types [50]. Analysis of blood protein polymorphism showed that Jakar and Sakten are close and differ from the Sibsoo type. Jakar and Sakten are similar to Mongolian Chinese Sheep, Bhyanglung, and Baruwal, while Sipsoo shows similarity to Baruwal [51].

Sheep are sheared three times a year with an average wool yield of 0.3–0.8 kg from purebred sheep while crossbred sheep can yield from 0.5 to 1.5 kg wool. Sheep are polyestrus, and their main mating season is March–April, June–July, and September–October. Usually, a single lamb is born, but twins are possible. Moreover, breeding rams are produced in the same flock. Due to inbreeding, there is low fertility, low prolificacy, and low resistance to diseases [49]. The reproductive performance of the sheep is provided in Table 9.

Domestic sheep are at risk as farmers abandon sheep husbandry practices that are no longer economically viable [49,52]. Farmers are also looking for alternative sources of revenue. This is evident from the population of sheep decline: within a decade there was a loss of 10% sheep population in Bhutan (Table 1). The sheep genetic resources are rich in Bhutan; therefore, it is important to selectively breed, conserve, and use sustainability.

### 2.6. Goat (Capra hircus)

Goats are popularly referred to a “Poor man’s cow” and provide nutrition to marginalized resource poor farmers with limited land holding and no other livestock [53]. Usually, the goat farming provides short economic return in quick time—from schooling of children to paying taxes. Farmers use them for meat, manure, religious offerings, and other economic purposes. Goats are offered as sacrificial animals to appease deities and as a meal for festivities such as social occasions of a new year and weddings [54]. They are popular among rural folks as the goats are able to utilize and survive on waste fodder such as *Artemisia,* and *Eupatorium* [55,56].

Bhutan’s goats predominantly found in Chukha, Samtse, Sarpang, Tsirang, and Dagana Dzongkhag (districts). They are similar to Bezoar type goats, and the goat native to Bhutan is considered to be the Black Bengal. It has predominantly black color with white or brown color infrequently. The horn is twisted, and the face is bearded; female adults weigh 20–25 kg, and male adults weigh 25–30 kg. Kidding twins are common, with triplets and quadruplets also being born [54]. The reproductive traits are presented in Table 10.

The morphological, biochemical, and mtDNA analysis showed no notable differences among local population, but they genetically differed from Indian or Tibetan goats [57]. Other breeds of goat are Jamunapari, Sirohi, Boer, and Beetal [54] which are not described here.

A slight increase in the goat population has been noted in the last decade (Table 1). Earlier, the number of goats reared was restricted to four, but the rule has been amended. This was mainly because goats destroy tender plants [53,58]. Goat rearing is easy and is shown to provide nutrients and food security to the farmers. Therefore, it is crucial to selectively breed and support technological innovation in goat farming including feed provision which can enhance and help farmers in utilizing the goat genetic resources effectively.

### 2.7. Chicken (Gallus domesticus)

Rural chickens are important in terms of providing economic opportunities to farmers. They provide eggs, meat, manure, and income. Moreover, they play a crucial role in socio-culture and the economy: native chickens are sacrificed to appease deities, honor guests, and restore women’s health during pregnancy through after-birth, and birds can also be provided as a gift [15,59,60].

Ten native chicken strains were characterized by Dorji et al. [61] and recognized mainly by phenotypic characteristics: Belochem, Baylaitey, Bobthra, Barred, Pulom, Yuebjha Kaap, Yuebjha Naap, Kauray, Naked neck, and Shekheni (Table 11). In addition, the qualitive traits of Bhutan’s chickens are studied by [62]. These studies revealed that the chickens are morphometrically diverse (Table 11). The genetic diversity study of traditional chicken types in Bhutan reported the origin as Red Jungle Fowl [60,63]. Moreover, exotic breeds have been introduced since 1960s to enhance egg and meat production in the country; these are the Rhode Island Red, Australorp, and White leghorn breeds which have no success as they are not adopted by the farmers. Additionally, grandparents of Keystone Browns, Hyline Brown, and Hyline Silver brown layers are imported to provide chicks and pullets to farmers for egg production while Ross308 and Cobb500 are two broiler breeds imported for meat purposes [58].

The indigenous chickens have low production performance (Table 12) with the length of lays ranging from 12 to 25 days averaging over 57 eggs per year. However, under an improved system of management, an average of 85 eggs per year was reported [64]. Therefore, there is an opportunity to selectively breed for higher production performance.

The Table 1 shows that, overall, the population of chickens is increasing as it represents both exotic and local chickens. However, it is not very well understood the extent to which the local population has been decreasing. Farmers rear anywhere from 6 to 19 chickens in a flock per household; however, the population of native chickens is declining with increased predation, diseases, shortages of feed, and shortages of labor. The decline in population is also attributed to farmers opting for exotic chickens for better production [58]. Moreover, there is no systematic breeding in rural chickens with available roosters used for breeding for all hens in a stock which leads to inbreeding depression. Therefore, it is important to select and initiate improvement of indigenous chicken in the country [64]. Therefore, there are lot of opportunities for sustainable utilization and conservation of these available chicken genetic resources, including research and development by taking cues from other countries in genetic improvement programs [65].

### 2.8. Buffalo (Bubalus bubalis)

Buffaloes in Bhutan are mostly non-descript riverine types with farmers identifying them as Kagay (entire body is black), Hyakulae (white or light gray stripes below the neck region), and Dobla (a cross between local and the Indian surti breed). Farmers use them for milk, meat, manure, and draught power with an average herd size of three [66]. They are found in Chukha, Samtse, Sarpang, Tsirang, and Dagana [67]. Reproductive and productive performance as reported by farmers are presented in Table 13.

Rearing buffaloes in Bhutan is challenging due to unavailability of quality breeding bull, drying up of wallowing pond, fodder scarcity, farmers preference of Jersey cattle, and inadequate veterinary support [58,66,67,68]. However, this also provides an opportunity for the government to support farmers in terms of an economic stimulus package as buffalo farming is seen to be highly lucrative due to a higher economic return and easier management than Thrabum [67]; this will in turn help in saving the AnGR in the country for sustainability and conservation.

### 2.9. Horse (Equus caballus)

Bhutan’s horses are predominantly Yuta (local) (97%) and Boeta (Tibet), Sharta (East, especially meaning Arunachal Pradesh), and Haflinger crosses, of which Yuta, Boeta, and Sharta are considered to be native to Bhutan [69]. While another horse type, Merak-Saktenpa, is considered [58], their distinctiveness is not proven [70]. Spiti, a Himachal Pradesh horse, is imported from India from 2000 onward, and they are maintained at the National Horse Breeding Farm, Bumthang [70]. The horses are mainly reared by the rural communities for transport of commodities and horticulture products and are usually available for hire by agencies in the government, contractors, and neighbors [70].

Reproductive performance is presented in Table 14. Yuta horse are surefooted, with a smaller body size, and have higher endurance than other types of horses [71]. A genetic variability study of traditional breeds revealed that Yuta and Boeta are similar while Sharta is differed from the earlier two [69]. Mules are produced by crossing Tibetan or Indian breeds of horses and donkey suggesting a morphometric similarity of Yuta and Sharta and their variation from Boeta (Tibetan) horses, and they are mainly used for carrying loads [70]. The population of the horses has declined by as much 37% (Table 1), and it is expected to decline as road networks and the transport system improves in Bhutan, and it has become imperative that the local pure line is conserved for sustainable utilization in future generations.

## 3. The Status of Animal Genetic Resource Conservation Programs

Cattle, yak, horse, and sheep have long been an integral element of Bhutan’s agricultural production system and economy. Traditional agricultural practices enabled cattle to be handled in a way that was environmentally friendly [72]. Crossbreeding with exotic breeds to boost productivity has put indigenous cattle breeds under peril as the human population grows and new economic requirements emerge [73]. The Royal Government of Bhutan (RGoB) prioritizes the protection and utilization of domestic animal genetic resources for sustainable livestock development in response to this danger and as a commitment to the Convention on Biological Diversity (CBD). Ex situ and in situ conservation efforts of domestic livestock biodiversity in Bhutan are being carried out by the Department of Livestock (DoL), the Council for Renewable Natural Resources (RNR) Research of Bhutan (CORRB), the College of Natural Resources (CNR), and the National Biodiversity Centre (NBC) [16]. Many conservation farms have recently been established to preserve local germplasm. Conservation initiatives, however, are insufficient unless suitable policies are in place.

In general, farmers are hesitant to raise animals. Farmers in Bhutan, for example, are rapidly abandoning sheep farming owing to declining economic returns, impacting the breeding of Jakar sheep, which are endemic to central Bhutan [49,50]. Ministry of Agriculture and Forest (MoAF) has responded by launching a participatory Jakar sheep breeding initiative in Phobjikha, central Bhutan, with the goal of encouraging Jakar sheep owners to continue farming sheep. Good breeding rams, free health care, wool processing technology, value addition, and easier marketing of sheep products are all examples of such incentives [16].

The RNR sector policy focuses on achieving greater national food security, protecting and managing natural resources, increasing rural income, and creating farm-based job possible [74]. Genetic evaluation and diversity studies for different animals have been initiated and performed for the following animals: cattle, Mithun, yaks, sheep, chickens, horses, and pigs. However, there is no elaborate genetic evaluation of traditional chicken types, donkeys, and goats. These genetic assessments can help conservationists and policymakers to frame a policy to protect and conserve animal genetic resources. Bhutan needs to study and classify breeds in a systematic way, as many breeds are currently unknown, and some will go extinct before they are documented.

The major FAnGR include cattle, yaks, pigs, chickens, horses, and buffalo. Except for goat and chicken, the population of traditional FAnGR has declined over the years; this decline is more dramatic in Mithun, buffalo, horses, pigs, and sheep. However, there is a dramatic increase in the chicken population, which can be attributed to commercial layer farming using Hy-Line Brown and related breeds for egg production rather than local breeds.

The genetic evaluation of the diversity of traditional cattle Nublang, yak, and sheep is completed, and Nublang is considered to be genetically unique and has been designated for conservation [2]. Moreover, many research and breeding centers have been established as of 2022 [42], promoting and conserving the local animal genetic resources in the country: the National Highland Research Development Centre, the Regional Native Poultry and Heifer Breeding Centre, the National Nublang Breeding Farm, the Regional Mithun Breeding Farm, the National Horse Breeding Farm, and the National Sheep Breeding Centre.

## 4. Conservation and Management

Conservation measures taken by countries encompass in situ and ex situ measures [75]. To conserve natural resources, Bhutan developed the first Biodiversity Action Plan (BAP) in 1998, a guiding policy document for the conservation and sustainable use of the biological resources of the country [16]. With a vision for “Effective Conservation, Sustainable utilization, and Equitable sharing of benefits arising from the access and use of biological resources” [76], initiatives for the conservation and management of FAnGR in Bhutan are entirely from RGoB through external funding support from UNDP-GEF (United Nations Development Program—Global Environment Facility) and the government of The Netherlands.

Most in situ conservation and management initiatives were supported through the Integrated Livestock and Crop Conservation Project (ILCCP) from the UNDP—GEF during 2007 to 2012 and implemented in collaboration with agencies of the ministry of agriculture and forest which includes District Livestock Sectors, the Department of Livestock, Central Livestock Farms, and Research Centers for Renewable Natural Resources (RNR) among others [2]. These projects have helped to kickstart many conservation farms, in situ as well as ex situ.

For in situ conservation, sustainable use through incentivization is advocated, whereas ex situ (those outside natural habit such as intensive farms and breeding centers) conservation farms are established for Belochem, Jakar sheep, Nublang, Mithun, native pigs, and horses. Moreover, the cryopreservation of germplasm for Nublang, Jituphab, chicken, and Jakar sheep is performed at the National Animal Gene Bank [2] at the National Biodiversity Centre (NBC); the ex situ conservation of preserved semen (gene banks) has been gradually evolving into a good national program.

In another project, the Agrobiodiversity Conservation Project, the government of The Netherlands has supported the establishment of the National Animal Gene Bank since 2005, which is a main focal department for looking after conservation of AnGR in the country [16]. Although the conservation and management of FAnGR have started, the established policies and rules need to be strengthened. Cryopreservation of the germplasm has started for some breeds of chicken and is expected to be rolled out to other breeds. Figure 2 shows the conservation and management of FAnGR in Bhutan. Some of the conservation efforts are the evaluation of genetic resources, education, the generation of awareness, the promotion of in situ conservation, and the promotion of traditional livestock rearing through the provision of incentives, as well as the formation of a groups. However, the policies need to be studied in terms of their effect on the attitude towards conservation of traditional breeds.

The National Animal Genebank (NAG) holds around 10,000 doses of sperm for local poultry, sheep, and pigs. Moreover, in the 11th five-year plan NBC collected DNA for Nublang, yak, and poultry breeds for development and utilization [16]. NBC offers “technical support for enhanced management methods, production, and marketing, education, and awareness on the importance of AnGR.” It also assesses, documents, and reports the genetic resources of domestic animals.

Communities in Sipsoo, Samtse; Merak in Trasigang; and Chumey in Bumthang are involved in on-farm conservation initiatives for sheep. Nublang has onsite conservation at Sombaykha, Haa. Saphak is conserved in Gomdar in Samdrup Jongkhar and Udzorong in Trashigang. Indigenous poultry is being conserved at Mendrelgang in Tsirang and breeds to produce a pure line of native poultry. NBC is also involved in improving animal husbandry practices, assisting in group formation, diversifying products, adding value to local breeds, and incentivizing and promoting traditional breeds. One such initiative is the establishment of the Nublang fund.

Genetic diversity studies have been carried out using conservation genetics tools, but the genetic and molecular techniques’ full potential needs to be seen in the country. The country is faced with a limited budget and requires more professionals capable of applying the latest technologies.

## 5. A Novel Program to Facilitate Genetic Resources in Bhutan

Genetic evaluation and molecular genetics are the two primary tools readily available for research in Bhutan’s conservation of animal genetics. Ongoing livestock data are available from the National Statistics Bureau and the National Biodiversity Center, along with publications such as Dorji and MacLeod [24], who estimated the genetic parameters of the Nublang (*Bos indicus*) of Bhutan. Based on Wangdi et al. [77] who studied the compositional quality of cow’s milk in Bhutan, animal genetic assessments will be likely be used in genetic studies to select animals in breeding programs. At the same time, research in molecular genetics appears to be advanced and more widespread than genetic evaluation, as evidenced by the increasing presence of biological laboratories in the NBC and at universities in Bhutan. Up to date study and research works on genetic diversity are conducted, and several marker genes associated with economically important traits have been studied, including the genetic structure of many livestock breeds. There have been studies on horses (*Equus caballus*) [69,70,71], cattle (*Bos indicus*) [19,20], indigenous chickens (*Gallus domesticus*) [60,61,62,65], sheep (*Ovis aries*) [50,51], and Mithun (*Bos frontalis*) [30,31,32,78,79], and research in molecular genetics can be a main tool in animal genetics, breeding, and conservation efforts in Bhutan. In the future, it is expected to see more in-depth studies in molecular genetics as a tool for conservation and sustainable utilization of farm animal genetic resources in Bhutan.

## 6. Conclusions and Recommendations

Local animal genetic resources are crucial for a country—for historical as well as cultural reasons—in addition to their economic utility. They are major part of food and nutrition security for the rural poor with the provision of milk, meat, manure, and draught power. They are also adapted to local climatic conditions and mostly are resistant to existing and emerging diseases.

National conservation efforts frequently place a high value on maintaining genetically different breeds. The government has started both in situ and ex situ conservation programs. For in situ conservation, education, and awareness, the promotion of conservation and sustainable use through incentivization is advocated and supported, whereas ex situ conservation farms have been established for Belochem, Jakar sheep, Nublang, Mithun, native pigs, and horses. These are invaluable genetic resources for Bhutan and the world. Moreover, cryopreservation of germplasm for Nublang, pigs, chicken, and Jakar sheep is performed at the National Animal Genebank, National Biodiversity Centre.

Bhutan needs to do more in terms of incentives for farmers to rear traditional breeds that are unique and adapted to the local environment. Additionally, a policy framework to conserve the genetic diversity, especially for AnGR, must be strengthened and promoted. Against this background, it is essential that the RGoB frames the policy to ensure that the genetic diversity is conserved which offers higher utility for generations to come. We need to balance between introducing new exotic breeds and preserving the native types, and molecular genetics and animal biotechnology can be crucial tools to preserve and promote the conservation of Farm Animal Genetic Resources (FAnGR) in Bhutan.

## Figures and Tables

**Figure 1 vetsci-10-00281-f001:**
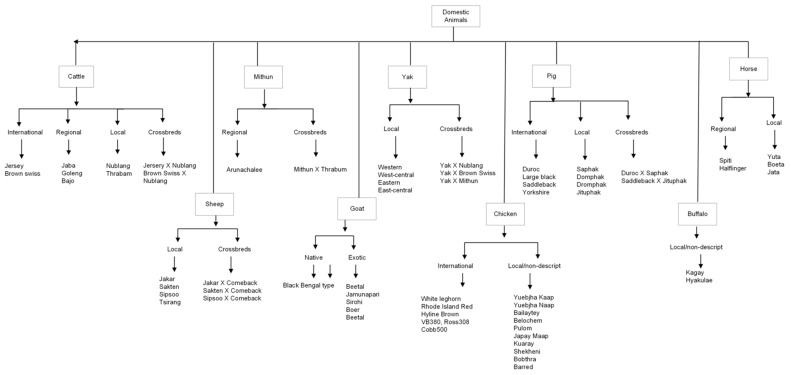
Species, types, and breeds of farm animal genetic resources in Bhutan.

**Figure 2 vetsci-10-00281-f002:**
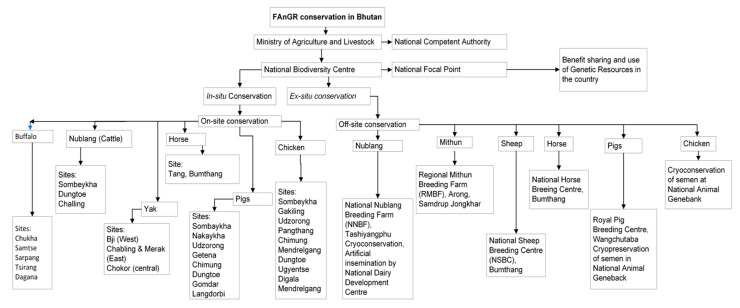
Farm Animal Genetic Resources and its conservation in Bhutan scheme.

**Table 1 vetsci-10-00281-t001:** Overview of Bhutan’s livestock population for the last decade and trends in population.

Breeds	Population Status (2010)	Population Status (2020)	Percentage Increase/Decrease
Cattle (Nublang)	309,277	281,015	−9.14
Mithun	571	390	−31.70
Yak	40,374	40,897	+1.30
Swine	19,711	17,577	−10.83
Sheep	12,699	10,793	−15.01
Goat	43,134	44,119	2.28
Chicken	349,004	1,383,714	+296.48
Buffalo	928	398	−57.11
Equine	23,423	14,649	−37.46

Data from National Statistics Bureau [18].

**Table 2 vetsci-10-00281-t002:** Descriptive of physical characteristics of Nublang.

Characteristics	Nublang (East)	Nublang (Center)	Nublang (South)	Nublang (West)
Wither height (cm)	104.5	110.5	110.1	110.1
Rump heigh (cm)	107.4	112.3	112.0	112.0
Heart girth (cm)	140.1	147.9	145.9	145.9
Paunch girth (cm)	140.2	149.8	149.3	149.3
Body length (cm)	120.1	127.4	124.3	124.3

Data from Tamang and Dorji [26].

**Table 3 vetsci-10-00281-t003:** Schematic representation of Mithun–Thrabam cross breeding strategy employed by farmers in Bhutan.

Parents		Progeny		Mithun Inheritance	Fertility of Progeny
Sire	Dam	Male	Female		
Mithun	Thrabam	Jatsha	Jatsham	50	Male: Sterile Female: Fertile
Nublang	Jatsham	Yangka	Yangkum	25	Male: Mostly sterile Female: Fertile
Nublang	Yangkum	Deob	Deobam	12.5	Male: Regaining fertility Female: Fertile
Nublang	Deobam	Deothra	Deothram	6.25	Male: Regaining fertility Female: Fertile
Nublang	Deothram	Nublang	Thrabam	3.125	Male: Fertile Female: Fertile

Data from Tamang and Dorji [26], Phangchung and Roden [27].

**Table 4 vetsci-10-00281-t004:** Physical characteristics measurements of adult yaks in three regions in Bhutan (mean ± SE).

Traits	Western		Central		Eastern	
	Male (*n* = 31)	Female (*n* = 32)	Male (*n* = 9)	Female (*n* = 19)	Male (*n* = 19)	Female (*n* = 12)
Height at withers (cm)	136 ± 1.2	117 ± 0.8	129 ± 1.9	110± 5.8	128 ± 2.0	113 ± 1.6
Body length (cm)	159 ± 1.7	137 ± 1.2	152 ± 4.0	133 ± 1.2	148 ± 1.7	133 ± 2.3
Chest girth (cm)	194 ± 1.7	165 ± 0.9	185 ± 3.5	165 ± 1.3	177 ± 2.8	160 ± 1.6
Body weight (kg)	419 ± 11.0	264 ± 4.0	369 ± 22.0	252 ± 4.5	323 ± 12.0	239 ± 8.0
Height at hips (cm)	107 ± 0.9	97 ± 2.2	101 ± 1.8	91 ± 0.9	101 ± 1.2	93 ± 1.7
Cannon circumference (cm)	21 ± 0.2	17 ± 0.2	20 ± 0.3	16 ± 0.2	19 ± 0.2	16 ± 0.2

Data from Dorji et al. [36].

**Table 5 vetsci-10-00281-t005:** Production and reproduction performance of different yak types in Bhutan (mean ± SE).

Traits	Western	Central	Eastern
Summer average daily milk (L)	1.19 ± 0.03	1.15 ± 0.04	1.27 ± 0.05
Winter average daily milk (L)	0.31 ± 0.16	0.30 ± 0.17	0.33 ± 0.15
Lactation length (days)	232.4 ± 3.05	226.6 ± 3.05	244.3 ± 4.58
Average lactation yield per female yak (L)	145.2 ± 25.00	140.3 ± 25.00	154.9 ± 20.00
Age at puberty (days)	1123.3 ± 29.60	1001.3 ± 32.60	1082.5 ± 29.90
Age at calving (days)	2646 ± 58.00	2540 ± 70.08	2529 ± 56.60
Calving interval (days)	484.6 ± 7.93	493.8 ± 9.76	480.9 ± 7.02
Gestation length (days)	257 ± 0.82	252.8 ± 0.57	256.6 ± 0.83
Services per conception (No.)	1.65 ± 0.09	1.49 ± 0.10	1.57 ± 0.09

Data from Wangdi and Wangchuk [40].

**Table 6 vetsci-10-00281-t006:** Descriptive characteristics of pigs found in Bhutan by regions.

Regions	Location	Features
Eastern	Mongar, Lhuntse, Tashigang, Pema Gatshel, Samdrup Jongkhar, Tashi Yangtse	Hairs: Sparse hairs Body: medium with bristles along the dorsal line Snout: medium Ears: medium prickly Tails: Curly to straight
East–Central	Sarpang, Zhemgang, Trongsa, Bumthang	Hair: sparse to medium density Body: medium Snout: Straight Ears: Medium sized prickly ears Tail: long and straight (Female has a sagging belly)
Western	Chukha, Thimphu, Haa, Paro, Samtse	Hairs: Sparse to dense (straight) Body: Medium Snout: short-medium, cylindrical Ear: medium size prick ears
West–Central	Gasa, Punakha, Wangdue, Dagana, Tsirang	Hair: dense Body: broad rectangular shape with bristle along the dorsal line (female with saggy belly) Snout: medium and cylindrical Ear: small-medium-large prick/droopy ears Tail: long and straight

Data from Nidup et al. [45].

**Table 7 vetsci-10-00281-t007:** Production traits of native pigs reared in rural areas.

Production Traits	Measurements
Piglet weight at birth (kg)	0.2–0.5
Total litter weight at birth (kg)	3
Total litter weight at weaning (kg)	22.5
Weaning weight (kg)	3–6
Adult live weight (kg)	50–100
Live weight at first service (gilt)	15–22
Age at first service (months)	5–12 (gilt); 5–9 (Boar)
Litter size (number)	3–9
Litter size–weaning (number)	2–8
Weaning age (days)	90
Farrowing/sow/year (number)	2
Productivity (stud boar) (year)	3–7

Data from Prasad and Sherpa [48].

**Table 8 vetsci-10-00281-t008:** Physical characteristics and traits of different sheep types in Bhutan.

Types	Features	Tracts	Adult Weight (kg)
Jakar	Body: Small Color: black, white, brown head and limbs Hair: Medium fine fleece Poll: female (poll), male (horns, thick horns run backwards, downwards, and twist forward) Nose: Roman	Temperate areas: Wangdue, Trongsa, Bumthang	25–30
Sakten	Body: medium Color: white/mixture of brown or black Hair: Finer coat Poll: both male and female have short and thin horns	Trashigang	35–40
Sipsoo	Body: tall Color: White/Patchy, black, brown head Hair: long and coarse Ear: short and tubular Litter: prolific/twins common	Chukha, Samtse, Tsirang, Dagana	60–70
Sarpang	Body: small, larger than Sipsoo Hair: coarse Color: white (most), brown Poll: female polled, male have horns	Sarpang	Not available

Data from Dorji and Tshering [49] and Dorji et al. [50].

**Table 9 vetsci-10-00281-t009:** Reproductive traits of sheep in Bhutan.

Traits	Age
Average age at 1st Service (months)	18
Average ate 1st Lambing (months)	23
Average lambing interval (months)	12
Reproduction rate	Low

Data from Dorji and Tshering [49].

**Table 10 vetsci-10-00281-t010:** Reproductive traits of indigenous goats in Bhutan.

Traits	Measurement
Age at puberty (months)	6.8 ± 0.12
Age at first kidding (months)	12.2 ± 0.14
Kidding interval (months)	6.5 ± 0.04
Kids per year (numbers)	2.5 ± 0.08
Life expectancy of Doe (years)	12.8 ± 0.23
Reproductive life of Doe (years)	9.8 ± 0.13
Total kids during lifetime (numbers)	19.6 ± 0.51

Data from Zangpo et al. [54], Tainang et al. [55], and Tamang et al. [56].

**Table 11 vetsci-10-00281-t011:** Distribution of Bhutanese Indigenous chickens in the country.

Chicken Strains	Distribution (Districts)
Belochem (crested)	Chukha, Dagana, Mongar, Pema Gatshel, Samtse, Sarpang, Tashigang, Tsirang, Wangdue Phodrang
Baylaitey (dwarf)	Sarpang, Samtse, Tsirang
Bobthra	Chukha, Dagana, Mongar, Pema Gatshel, Samtse, Sarpang, Tashigang, Tsirang, Wangdue Phodrang
Barred	Chukha, Daga, Mongar, Pema Gatshel, Samtse, Sarpang, Tashigang, Wangdue
Yuebjha Kaap (white)	Chukha, Dagana, Mongar, Pema Gatshel, Samtse, Sarpang, Tashigang, Tsirang, Wangdue
Pulom (frizzle)	Dagana, Sarpang, Tsirang, Wangdue Phodrang
Kauray	Tsirang, Dagana, and Sarpang
Yuebjha Naap (black)	Chukha, Dagana, Mongar, Pema Gatshel, Samtse, Sarpang, Tashigang, Tsirang, Wangdue Phodrang
Khuilay (naked neck)	Chukha, Dagana, Mongar, Sarpang, Tsirang, Wangdue Phodrang
Shekheni	Dagana, Sarpang, Tsirang

Data from Dorji et al. [61].

**Table 12 vetsci-10-00281-t012:** Production performance of native chickens under scavenging condition (range).

Traits	Measurements
Egg production/hen/year (numbers)	60–85
Age at first egg (days)	182–210
Length of laying period/clutch (days)	12–25
Clutches/year	2–4
Eggs/clutch	15–20

Data from Tamang et al. [64].

**Table 13 vetsci-10-00281-t013:** Reproductive and productive performance (mean ± SD, and range) of Bhutan’s Buffaloes.

Traits	Measurements
Age at first calving (months)	42 ± 50
Gestation length (days)	313 ± 4.59
Calving interval (days)	540 ± 97.46
Lactation length (days)	315 ± 28.28
Productive life (years)	14.3 ± 2.46
Life span (years)	21 ± 3.36
Number of calving (lifetime)	10.4 ± 1.30
Milk yield (kilograms)	0.25–4.50
Age at first service (months)	18–50
Service per conception (numbers)	1–4

Data fromTamang et al. [67] and Wangdi et al. [68].

**Table 14 vetsci-10-00281-t014:** Reproductive performance of native horses in Bhutan.

Traits	Measurements
Age at sexual maturity (days)	728
Gestation length (days)	334–345
Inter Foaling Period (years)	1–2

Data from Dorji et al. [70].

## Data Availability

The data presented in this study are available upon reasonable request from the Network Center for Animal Breeding and Omics Research, Faculty of Agriculture, Khon Kaen University, Thailand.

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
