# Peer review of "Current Status and Conservation Management of Farm Animal Genetic Resources in Bhutan"

_vetsci, 2023, doi:10.3390/vetsci10040281_

Round 1

Reviewer 1 Report (Previous Reviewer 3)

In figure 2 there is used Nublang - what about other species of Bovides?

In paper there are cattle, mithum, yak and buffalo - four species of Bovides - please add Latin names of species on the beginning and repeat for time to time because not all names are common and buffalo is used also for Bison bison. In some places there are latin names but not for all species.

The latin names must be in italic - in the bibliography there are many of them

In table 1 plese change the proportion of two species into one number - the second is easy to calculate 

Also in Thrabam has 3.125% of Mithum - the crosbreds between Mithum and Thrabam shoul have 51.56% of Mithim (?)

Author Response

Dear reviewer

   We are very grateful for the critical reading and your efforts to improve the quality of the manuscript.  We hope the responses to each comment as listed below will please you.

Response to Reviewer 1 Comments

Point 1: In figure 2 there is used Nublang - what about other species of Bovides?

Response 1: Nublang is the traditional cattle breed of Bhutan. It supposes using “cattle” instead of “Nublang” Other bovines (buffalo and bison) are also provided in the revised Figure 2.

Point 2: In paper there are cattle, mithum, yak and buffalo - four species of Bovides - please add Latin names of species on the beginning and repeat for time to time because not all names are common and buffalo is used also for Bison bison. In some places there are latin names but not for all species.

Response 2: The Latin names for these species are included to differentiate the different bovines. See Lines 67-71, 118, 126,128, 144, 192, 251, 298, 342, 377, 407, 451, 471, 601, 609-610.

Point 3: The latin names must be in italic - in the bibliography there are many of them

Response 3: The changes have been made to italic for Latin names for both in-text and bibliography.

Point 4: In table 1 please change the proportion of two species into one number - the second is easy to calculate 

Response 4: The proportion is changed to represent only one number. See Table 1.

Point 5: Also in Thrabam has 3.125% of Mithum - the crosbreds between Mithum and Thrabam should have 51.56% of Mithim (?)

Response 5: Your question is very interesting, the blood level of Mithum and Thrabam should have 51.56% of Mithun if we use F5 generation of Thrabam to directly breed with Mithun. But in my opinion, before breeding Thrabam with Mithun again it is necessary to raise Thrabam's bloodline to equal or as close to purebred (100%) as possible using the upgrading mating system between Nublang and Thrabam However, detailed blood analysis in animals is still difficult due to the lack of accurate equipment.

Best Regards,

Wuttigrai Boonkum

Reviewer 2 Report (Previous Reviewer 2)

The revised manuscript shows several improvements. However, it is very hard to track changes with reference to each comment given there are very few sections where the revised text has been highlighted. There are a few unaddressed comments as well, For example (FAO - line 33) and spelling checks for Gaur vs Guar. Therefore, it is suggested to refer to the comments to improve this manuscript.

Author Response

Dear reviewer

   We are very grateful for the critical reading and your efforts to improve the quality of the manuscript.  We hope the responses to each comment as listed below will please you.

Response to Reviewer 2 Comments

Point 1: The revised manuscript shows several improvements. However, it is very hard to track changes with reference to each comment given there are very few sections where the revised text has been highlighted. There are a few unaddressed comments as well, For example (FAO - line 33) and spelling checks for Gaur vs Guar. Therefore, it is suggested to refer to the comments to improve this manuscript.

Response 1:        Line 33. FAO full form has been provided.

Lines 202 – 206: Gaur is consistently now used.

An earlier version, there were no track changes, so it is hard to track the changes. However, following answers were provided for the comments to an earlier review. The questions and answers are marked red, while the answer is highlighted in yellow.

The manuscript is very well written and provides a nice overview of the Livestock and its conservation status in Bhutan. There are a few areas to improve, mainly 2 major suggestions and then a few minor corrections.

First, there should be some information to highlight how those breeds are different in terms of genetic relatedness, production, body features etc., or they can be just another colour variant of a nearby local population only separated by the mountain terrain of Bhutan.

Answer: More information has been provided in the content of each species.

Second, a couple of country maps with each species (or group a few small species, e.g., goat and sheep) to show location of important breeds.

Answer: A country map has yet to be provided, but the location is described within the content of each species.

 L16: It is not clear that these “livestock breeds” belong to which species.

Answer: This has been made clear to revise the sentence as “Some unique livestock breeds in Bhutan are Nublang (cattle breed), Yak, Sapha (pig breed), Yuta (horse breed), Merak-Saktenpata (horse breed), and Belochem (chicken breed).” See lines 22-24.

L17: The drop was within a decade or century? And also quantify (%) drop.

Answer: Drop within the decade. The percentage is mentioned in the Table 1.

L26: FAO?

Answer: Full form provided in line 33

L27: “exist on this” ?

Answer: We already removed this word from revised manuscript.

L27: only 38 breeds ? it should be much higher in the global context.

Answer: We revised the sentences as “8,800 different livestock breeds exist—which is often referred as “global animal genetic resources (AnGR)” with about 38 different species to meet our needs for food, clothing, draft power, and saving money [1]. These animal species, breeds, and strains have economic, scientific, and cultural values to mankind for food and agricultural production [2,3] playing a crucial role in global food systems as the main source of animal protein and one third of all protein consumed by people comes from animal-source foods [4]. Especially, native breeds are well adapted to local agro-ecological zones and resistant to emerging diseases [5] and provide sustained economic returns albeit small.” See lines 34-42.

L102: “crossed” to “cross-bred”. Please include if you know the proportion (e.g., 50:50, or less) of each exotic breed in overall population related to the Jersey and Brown Swiss.

Answer: Change crossed to crossbred in all place in revised manuscript. The proportion of exotic to local is included in the table.

L119: Please include the scientific name of “gaur” and be consistent with using either “gaur” or “Gaur” throughout the manuscript.

Answer: Consistently changed to Gaur.

L124: Clarify that “Nublang” is a cattle. Also provide some information if the hybrids are fertile, both from Yak and Mithun.

Answer: The Nublang is the male native cattle, while Thrabum is the female counterpart. However, in Bhutan, the male counterpart is known as Nublang, which is considered to be an important genetic resource in the country. That’s why the section title has been kept as Nublang. The information about hybrids are provided in table 2.

L232-247: These abbreviations are not required, most of them are not used and a couple of them have been used again with full form. So, it is better to remove them, please also check throughout the manuscript if any abbreviation is not being frequently re-used remove it.

Answer: This has been done.

L237 & 248:  What is RNR?

Answer:  Renewable natural resources. Full form is provided.

L276-281: UNDP-GEF and RNRRDCs?

Answer:  in Bhutan are entirely from RGoB through external funding support from UNDP-GEF (United Nations Development Programme – Global Environment Faciliy) and the government of the Netherlands

RNRRDCs is removed.

Finally, there are a lot of self-citations, which is not very inappropriate. However, if several of the articles from authors refer to the same/similar background or supporting information, please use only the most relevant .

Answer: There is one self-citation on using genetic markers to improve animal production.

Best Regards,

Wuttigrai Boonkum

Reviewer 3 Report (New Reviewer)

The authors of the article provide a review regarding the status of rare farm animal breeds in Bhutan and current attempts in this country to conserve Farm Animal Genetic Resources (FAnGR).

It may seem that the interest and usefulness of the study are limited by its narrow context. However, it is my view that this very circumscription of context makes this work more interesting and important, as it provides an interesting case study to measure the basic premise of any case we can make in favor of the conservation of farm animal breeds: their importance in a specific context.

The loss of diversity when it comes to the indigenous breeds of farm animals is real; the key point is to understand whether this constitutes an urgent emergency to rectify, or if it is instead a good thing, a sign of development and modernization. To answer this, it is necessary to understand the actual importance of indigenous breeds within their original range. This article analyzes the problem in my opinion quite effectively concerning the context of Bhutan, a country with characteristics particular enough to justify the authors' initial concerns about the gradual replacement of local breeds by those most widely used in industrial agriculture. Another merit of the article is that it also included in the assessment the cultural importance of a breed, and not solely its economic importance.

What I would ask the authors to add to the final draft of the paper, is to further explore the aspect of the value of indigenous breeds versus the alternatives employed in global industrial breeding. Indeed, in some cases, it seems to me that not enough arguments are provided to justify the protection of these lineages. For instance, too few reasons are provided why the Bhutanese should prefer the indigenous pig breeds to the fast-growing and bigger imported breeds. The same issue is present also when it comes to discuss chickens. Moreover, buffalos are included in the discussion, but their numbers shown in table 2 do not seem to justify inclusion (while also Mithun are few, their cultural importance as it is described in the text justifies the inclusion in the review).

Two more comments:

The word “livestock”—The concept of livestock can be confusing because it is usually used to refer to farm animals except for poultry. The article also deals with chickens, so I would avoid using this term unless providing a clear definition from the outset.

Lines 36-43—“These animal species, breeds, and strains have economic, scientific, and cultural values to mankind for food and agricultural production [2,3] playing a crucial role in global food systems as the main source of animal protein and one third of all protein consumed by people comes from animal-source foods [4] livestock farming at a housed, or a semi-commercial level, provides proof farmers with not only their daily sustenance but also for draught power and transport which in turn fulfills the diverse needs of the farmers, which “improved breeds” otherwise couldn’t provide”.

This sentence is rather complicated and confusing. It may get to the reader the idea that livestock (the term used before) accounts for the main source of animal protein. It probably is, according to many estimates, but its consumption is not much more spread than that of poultry. Moreover, it is hard to understand if the subject here includes all kinds of livestock, including widespread commercial breeds, or only rare, local breeds. In the first case, the sentence is not very informative concerning the topic of the paper. In the second case, it is not the case.

Author Response

Dear reviewer

   We are very grateful for the critical reading and your efforts to improve the quality of the manuscript.  We hope the responses to each comment as listed below will please you.

Response to Reviewer 3 Comments

Point 1: The authors of the article provide a review regarding the status of rare farm animal breeds in Bhutan and current attempts in this country to conserve Farm Animal Genetic Resources (FAnGR).

It may seem that the interest and usefulness of the study are limited by its narrow context. However, it is my view that this very circumscription of context makes this work more interesting and important, as it provides an interesting case study to measure the basic premise of any case we can make in favor of the conservation of farm animal breeds: their importance in a specific context.

The loss of diversity when it comes to the indigenous breeds of farm animals is real; the key point is to understand whether this constitutes an urgent emergency to rectify, or if it is instead a good thing, a sign of development and modernization. To answer this, it is necessary to understand the actual importance of indigenous breeds within their original range. This article analyzes the problem in my opinion quite effectively concerning the context of Bhutan, a country with characteristics particular enough to justify the authors' initial concerns about the gradual replacement of local breeds by those most widely used in industrial agriculture. Another merit of the article is that it also included in the assessment the cultural importance of a breed, and not solely its economic importance.

What I would ask the authors to add to the final draft of the paper, is to further explore the aspect of the value of indigenous breeds versus the alternatives employed in global industrial breeding. Indeed, in some cases, it seems to me that not enough arguments are provided to justify the protection of these lineages. For instance, too few reasons are provided why the Bhutanese should prefer the indigenous pig breeds to the fast-growing and bigger imported breeds. The same issue is present also when it comes to discuss chickens. Moreover, buffalos are included in the discussion, but their numbers shown in table 2 do not seem to justify inclusion (while also Mithun are few, their cultural importance as it is described in the text justifies the inclusion in the review).

Response 1: The comment is helpful, but still trying to figure out what to include in the value of indigenous breeds versus global industrial breeding. One is biodiversity, which has been highlighted in the text. Second, the value of indigenous species is derived from social, cultural, and economic benefits. Most of the section covers indigenous breeds' contribution to farmers' sustenance. However, it would be difficult to say what would be the reason for farmer farmers to pick up indigenous pig rearing rather than exotic. Probably, it will be useful to incentivize or provide subsidies. Therefore, I have added a line in the text: “Moreover, providing incentives or subsidy through policy intervention might help farmers to rear native pigs helping its conservation.” See Lines 339 – 340.

Similarly, the chickens’ egg production and growth characteristics are low, and farmers do not want to rear. Therefore, it will only be incentivizing or subsidizing that might help through policy interventions. Culturally native chickens are used as follows: “native chickens are sacrificed to appease deities, honor guests, restore women’s health during pregnancy through after-birth, and birds can also be provided as a gift”.  See Lines 410 – 412.

Buffaloes are included here because only a few studies have been done on Buffalo in Bhutan. Therefore, it would be useful for readers to gain information about Bhutanese buffaloes because the data is scarce. We would remove it if it doesn’t add value to the paper, but for now, we have decided to keep it.

Two more comments:

Point 2: The word “livestock”—The concept of livestock can be confusing because it is usually used to refer to farm animals except for poultry. The article also deals with chickens, so I would avoid using this term unless providing a clear definition from the outset.

Response 2: Following sentence has been added to clear the confusion for the purpose of this review: Lines 99-100: For this particular review, livestock refers to farmed domestic animals including poultry, for convenience of discussion.

Point 3: Lines 36-43—“These animal species, breeds, and strains have economic, scientific, and cultural values to mankind for food and agricultural production [2,3] playing a crucial role in global food systems as the main source of animal protein and one third of all protein consumed by people comes from animal-source foods [4] livestock farming at a housed, or a semi-commercial level, provides proof farmers with not only their daily sustenance but also for draught power and transport which in turn fulfills the diverse needs of the farmers, which “improved breeds” otherwise couldn’t provide”.

This sentence needs to be simplified and clarified. It may get to the reader the idea that livestock (the term used before) accounts for the main source of animal protein. It probably is, according to many estimates, but its consumption is similar to that of poultry. Moreover, it is hard to understand if the subject here includes all kinds of livestock, including widespread commercial breeds, or only rare, local breeds. In the first case, the sentence concerning the paper’s topic is not very informative. In the second case, it is not the case.

Response 3: The sentence is changed to include the following:

 “These animal species, breeds, and strains have economic, scientific, and cultural values to mankind for food and agricultural production [1, 2] playing a crucial role in global food systems as the main source of animal protein and one-third of all protein consumed by people comes from animal-source foods [4]. Especially, native breeds are well adapted to local agroecological zones and resistant to emerging diseases [3] and provide sustained economic returns albeit small.” See line 37 – 42:

Other changes made:

Table numbers are updated.

All the farm animals’ Latin names are provided and highlighted.

Best Regards,

Wuttigrai Boonkum

This manuscript is a resubmission of an earlier submission. The following is a list of the peer review reports and author responses from that submission.

Round 1

Reviewer 1 Report

‘Conservation of Farm Animal Genetic Resources (FAnGR) in Bhutan’ attempts to summarize the status of FAnGR and the efforts to conserve them. However, this manuscript is not well summarized.

1. The status quo and measures of other species conservation can be introduced in the introduction, rather than just emphasizing the importance of preservation of indigenous breeds.

2. The introduction should be logical and coherent, rather than simply piecing together sentences.

3. The author did not thoroughly analyze the causes of genetic resource loss of each breed, but included too many uncertain factors, such as ‘we are not sure the if the relative loss of native pig is higher’, ‘Furthermore, due to lower productivity relative to exotic breeds, many farmers choose and grow enhanced or exotic breeds, for which statistics are limited and for which there is no systematic census.’ and ‘This is probably due to import of exotic breeds of goat from India—however, confirmation is required.’

Reviewer 2 Report

The manuscript is very well written and provides a nice overview of the Livestock and its conservation status in Bhutan. There are a few areas to improve, mainly 2 major suggestions and then a few minor corrections.

First, there should be some information to highlight how those breeds are different in terms of genetic relatedness, production, body features etc., or they can be just another colour variant of a nearby local population only separated by the mountain terrain of Bhutan.

Second, a couple of country maps with each species (or group a few small species, e.g., goat and sheep) to show location of important breeds.

L16: It is not clear that these “livestock breeds” belong to which species.

L17: The drop was within a decade or century? And also quantify (%) drop.

L26: FAO?

L27: “exist on this” ?

L27: only 38 breeds ? it should be much higher in the global context.

L102: “crossed” to “cross-bred”. Please include if you know the proportion (e.g., 50:50, or less) of each exotic breed in overall population related to the Jersey and Brown Swiss.

L119: Please include the scientific name of “gaur” and be consistent with using either “gaur” or “Gaur” throughout the manuscript.

L124: Clarify that “Nublang” is a cattle. Also provide some information if the hybrids are fertile, both from Yak and Mithun.

L232-247: These abbreviations are not required, most of them are not used and a couple of them have been used again with full form. So, it is better to remove them, please also check throughout the manuscript if any abbreviation is not being frequently re-used remove it.

L237 & 248:  What is RNR?

L276-281: UNDP-GEF and RNRRDCs?

Finally, there are a lot of self-citations, which is not very inappropriate. However, if several of the articles from authors refer to the same/similar background or supporting information, please use only the most relevant .

Reviewer 3 Report

many sentences are unclear and this shortcuts need to be explained - for exaple line 14 "farmers are rearing livestock that narrow the renge of genetic diversity". I understand that farmers are using more productive breeds, so the population of native breeds is decreasing and in consequence the genetic diversity too. But genetic diversity within or between breeds?

line 27 - 38 domectic breeds from 8800 meet our needs? -

The rest of introduction chapter, especially last two paragraphs are a little mess. There is very unclear presentation of ideas and facts. f.e. what kind of tool is used by goverment to force crossbreeding, How the  Constitution statments are implemented, in situ or only ex situ conservation is important. 

The description of local breeds is also unclear for samebody who do not know the geography of Bhutan could be lost. 

In table 2 there are size of global population - what is the size of local breeds population? The decrease of global population of cattle can or cannot be related to local breeds stuation and genetic diversity level.

Jatsa and Jatsham are not breeds (line 125)

line 201 - whos "current strategy" and how this sentence is connected with establishing preservation farm for some native breeds?

The chapter no 3 is general and not connencted with Bhutan situation - can be in Introduction part.

What means "in situ education"? (line 283)

Why the alive animals kept in farm are called ex situ? Are they kept out of the region? 

In chapter 5 there are information about project and activity of different institution but there is a lack of details - when, how large, e.g. how large is nucleus herd - what is the method of selection and mating there?  The chapter seems to be information about plans not about results (project started more than 10 year ago), so some should be known

So the whole paper is very general, with no detailed information about status and situation of local breeds, with no information about results of longterm projects. For average reader this paper is not informative at all. Also the paper is not prepared in a way required for scientific paper. 

In conclusion, in my opinion, the structured information about native breeds could be presented as scientific paper but the actual version full of wishes and general information,  has to be rejected.